# No universal mathematical model for thermal performance curves across traits and taxonomic groups

Dimitrios - Georgios Kontopoulos ●[1,2,3] ✉, Arnaud Sentis[4], Martin Daufresne[4], Natalia Glazman ●[1], Anthony I. Dell[5,6] & Samraat Pawar ●[1]

In ectotherms, the performance of physiological, ecological and life-history traits universally increases with temperature to a maximum before decreasing again. Identifying the most appropriate thermal performance model for a specific trait type has broad applications, from metabolic modelling at the cellular level to forecasting the effects of climate change on population, ecosystem and disease transmission dynamics. To date, numerous mathematical models have been designed, but a thorough comparison among them is lacking. In particular, we do not know if certain models consistently outperform others and how factors such as sampling resolution and trait or organismal identity influence model performance. To fill this knowledge gap, we compile 2,739 thermal performance datasets from diverse traits and taxa, to which we fit a comprehensive set of 83 existing mathematical models. We detect remarkable variation in model performance that is not primarily driven by sampling resolution, trait type, or taxonomic information. Our results reveal a surprising lack of well-defined scenarios in which certain models are more appropriate than others. To aid researchers in selecting the appropriate set of models for any given dataset or research objective, we derive a classification of the 83 models based on the average similarity of their fits.

All physiological, ecological and life history traits of ectotherms, from cellular metabolic rates to population growth rates and species interactions, are strongly influenced by temperature. The relationship between trait performance and temperature is known as the "thermal performance curve" (TPC) or the "thermal reaction norm" (Fig. 1a) and is unimodal and usually asymmetric[1,2]. Determining the most appropriate TPC model for a given trait dataset and how the shape of the TPC varies across trait types, taxonomic groups, and environments have a wide range of applications across biological systems and levels of organisation, from cellular metabolic modelling[3–5] and ontogenetic growth[6–8], to population[9–12], community[13–15], ecosystem[16–18], and disease dynamics[19–21]. TPCs have also recently become integral to predicting the effects of climatic warming as well as thermal fluctuations on biological systems[22–25].

A wide variety of unimodal TPC models have been developed since the first two (one symmetric and one asymmetric) were proposed by Janisch back in 1925[26]. TPC models span the spectrum of phenomenological (e.g., based on a modification of the Gaussian distribution[27,28]) to mechanistic (based on biochemical kinetics[29–33]) mathematical equations. Many of the phenomenological models were initially developed for specific traits or species groups (e.g., for the development rate of arthropods[34–36]).

This smorgasbord of TPC models prompts the question of whether certain models are more appropriate than others for specific trait

[1]Department of Life Sciences, Imperial College London, Silwood Park, Ascot, Berkshire, UK. [2]LOEWE Centre for Translational Biodiversity Genomics, Frankfurt, Germany. [3]Senckenberg Research Institute, Frankfurt, Germany. [4]INRAE, Aix Marseille University, UMR RECOVER, Aix-en-Provence Cedex 5, France. [5]National Great Rivers Research and Education Center, East Alton, Illinois, USA. [6]Department of Biology, Washington University in St. Louis, St. Louis, Missouri, USA. ✉e-mail: dgkontopoulos@gmail.com

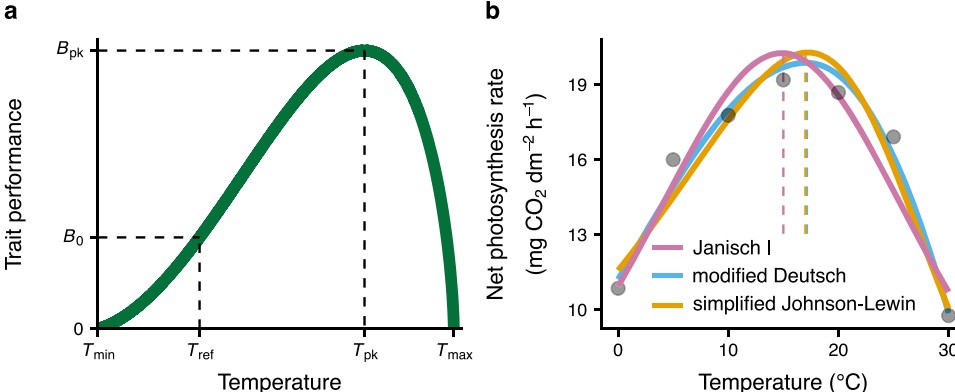

**Fig. 1 | The thermal performance curve (TPC) illustrates the influence of temperature on the performance of a biological trait of an individual ectotherm, a population, or even a whole ecosystem.** TPCs can be estimated by fitting a wide range of nonlinear mathematical models to trait measurements obtained at multiple temperatures. Some parameters commonly included in such models are explicitly shown in panel (**a**). $T_{min}$ and $T_{max}$ are the temperatures above and below which trait values are positive. $T_{pk}$ is the temperature at which the TPC reaches its maximum height ($B_{pk}$). $B_0$ is the performance at a reference temperature ($T_{ref}$) below $T_{pk}$, often quantified for comparison of baseline performance across TPCs from different individuals or species[63]. Panel (**b**) shows three different models (see Supplementary Note 7) fitted to net photosynthesis rate measurements of the eastern daisy fleabane flowering plant (*Erigeron annuus*)[79]. Note that the resulting TPCs differ in their $T_{pk}$ (dashed lines) and $B_{pk}$ values, as well as in their degree of skewness, with the TPC of the Janisch I model being fully symmetric. Source data for panel (**b**) are provided as a Source Data file.

data. For example, mechanistic models derived from biochemical kinetic principles were most often developed under the assumption that a single rate-limiting enzyme governs the shape of the TPC[29–33,37,38]. Because of this, such models are commonly assumed to strongly outperform phenomenological models for fundamental physiological traits (e.g., respiration, photosynthesis). In contrast, the TPCs of more "emergent" traits, such as overall dynamic body acceleration, resource consumption rate, and growth rate, cannot be straightforwardly linked to the activity of a single rate-limiting enzyme and are expected to be better described by phenomenological models whose parameters more accurately capture various features of TPC shape (e.g., $T_{min}$, $T_{pk}$, and $T_{max}$; Fig. 1a). It is worth noting that, even for fundamental physiological traits such as metabolic rate, the aforementioned assumption of a single rate-limiting enzyme is an approximation[39–42]. Another assumption is that for high-resolution datasets (numerous temperature treatments over a sufficiently wide range[43]), parameter-rich models will systematically outperform simple alternatives, e.g., revealing two different gradients at the rising part of the TPC[43–45]. Such fine-scale variation in TPC shape is assumed to be generally present but hard to detect because most thermal performance datasets tend to be small[46].

Whether such assumptions are valid has yet to be objectively and systematically determined, with all previous such studies being limited in their scope in terms of the number and types of models compared, types of traits, and diversity of taxonomic groups[27,47–52] (Supplementary Table 1 in Supplementary Note 1). Even in recent studies that introduce new general models (expected to apply well to wide classes of datasets)[31–33,53], a thorough performance comparison between the new model and pre-existing ones is typically missing. As a result, in studies quantifying the temperature dependence of experimental trait data, TPC models are typically chosen (semi-)arbitrarily and with limited justification. This practice could occasionally worsen the signal-to-noise ratio of a dataset or, worse, introduce model-specific biases that could be hard to control for[51] (see Fig. 1b). For example, consider a hypothetical trait whose TPC is strictly symmetric. For such a trait, models that force the fall of the TPC to be steeper than its rise will necessarily yield misleading estimates of the gradients before and after the thermal optimum. Furthermore, some datasets may not allow adequate statistical power to objectively estimate all parameters of a complex and parameter-rich model ("parameter unidentifiability"), in

which case multiple sets of parameter estimates will produce effectively identical curves. One could treat this spurious variation in parameter estimates as real rather than a purely statistical artefact and may attempt to come up with mechanistic explanations for it.

To fill this gap, here we compile an extensive set of TPC models that covers practically all models that have been proposed to date, which we fit to multiple experimentally-determined thermal performance datasets from the literature. We then compare models on the basis of (i) how well they fit experimental data and (ii) how similar their fits tend to be. This allows us to address three key questions:

1. Are there models that consistently outperform others across any dataset?
2. Do models with more parameters tend to outperform simpler alternatives as the sampling resolution of the dataset increases?
3. Do mechanistic models fit fundamental physiological traits better than phenomenological models?

## Results and discussion
### Thermal performance datasets analysed in this study
We compiled 3598 previously published datasets of thermal performance (i.e., measurements of trait performance vs temperature for a single taxon from a specific location and study), with at least four distinct temperatures per dataset (see "Methods"). These were filtered down to 2739 datasets[54], spanning more than 100 traits from all seven kingdoms[55] and from 39 phyla (Fig. 2). For almost all datasets, trait performance measurements were obtained under constant rather than fluctuating temperatures, even though the latter should offer a closer approximation to the conditions naturally experienced by a given species[56]. Even for datasets with constant temperatures, the time allowed for acclimation to each experimental temperature[57] was often not reported explicitly. We should emphasise here that the aforementioned data gaps did not prevent us from addressing the three key questions of the present study. Nevertheless, future studies could investigate whether the timescale of temperature shifts systematically influences (a) the shape of the TPCs of diverse traits and taxa and (b) the performance of alternative mathematical models.

### Comparison of model performance across datasets
We collected published TPC models that capture the entire TPC (i.e., not just its rise or fall) using Google Scholar (see "Methods"). We did

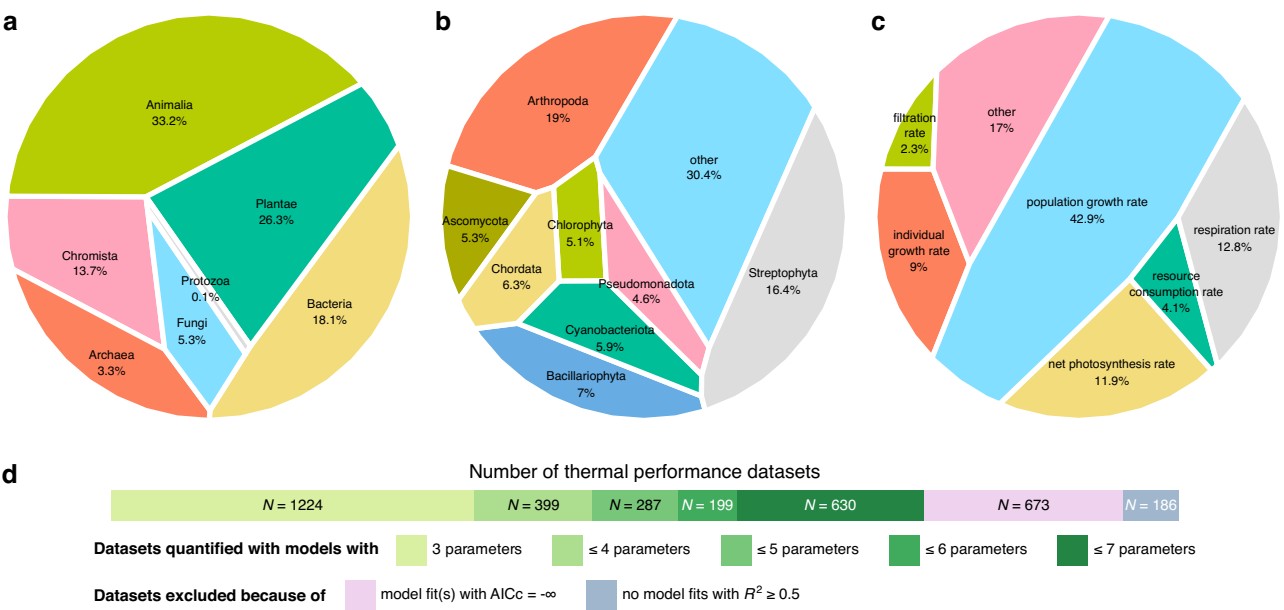

**Fig. 2 | Characteristics of thermal performance datasets included in this study.** Panels (**a**, **b**, and **c**) show the breakdown of the most common kingdoms, phyla, and traits, respectively, in our data compilation. Panel (**d**) shows the number of thermal performance datasets to which various models could be fitted, as well as datasets that were excluded from this study. For more details, see the Methods section. Source data are provided as a Source Data file.

# BOX 1:

# Model performance metrics used in this study

- **$R^2$:** A measure of the absolute performance of a model against a given dataset. It represents the proportion of variance in the response variable that is explained by the model. $R^2$ can take values between 0 and 1, albeit negative values can occur in some cases. Adding extra parameters to a given model generally increases the $R^2$ but can lead to overfitting, i.e., treating the noise in the data as true signal. When overfitting occurs, the model will reach a very high $R^2$ against the data on which it was trained, but much lower $R^2$ values against new observations.
- **AICc:** A measure of the relative performance of alternative models against a given dataset. It penalises models with large numbers of parameters and, therefore, can be used to avoid cases of overfitting. AICc values can range from $-\infty$ to $\infty$, and when comparing alternative models, the most appropriate model for a given dataset would be the one with the lowest AICc value. Note that AICc values are specific to a given dataset. For objectively comparing the performance of different models across multiple datasets, AICc weights can be used instead.
- **AICc weights:** A standardised measure of the relative performance of alternative models against a given dataset. AICc weights take values between 0 and 1, with their sum being equal to 1. In the case where a single model vastly outperforms all fitted alternatives, that model will have an AICc weight of $\approx 1$, whereas all other models will have AICc weights of $\approx 0$. At the opposite extreme, if all fitted models perform equally well against a dataset, their AICc weights will all be equal to 1 divided by the model count. It is worth stressing that, in contrast to the $R^2$, AICc weights cannot be used to evaluate the absolute performance of a model against a dataset. This makes AICc weights meaningless in the case where no model can fit the data well enough. For this reason, an $R^2$ cutoff (e.g., 0.5 in the present study) is typically used to exclude datasets that are poorly fitted by all candidate models.

not include models that can only be fitted to multiple thermal performance datasets at the same time (e.g., through a hierarchical Bayesian approach[53]) and not to individual datasets. We also ensured that each included model could adequately fit (see "Methods") at least a single dataset from a representative subset of our data compilation. This process yielded 83 distinct TPC models (see Supplementary Note 7 for their equations, free parameters, and references).

We fitted the 83 models separately to each thermal performance dataset and, for each fit, calculated the value of $R^2$ and Akaike Information Criterion, using the correction for small sample size (AICc[58]; see Box 1). After removing model fits with an $R^2$ value below 0.5 (see Supplementary Figs. 1 and 2 in Supplementary Note 2 for information on retained model fits), we calculated the AICc weight of each model for each thermal performance dataset. We also constructed a matrix of median pairwise Euclidean distances among models, from which we

inferred a dendrogram of models to identify those that generally produce very similar fits (Fig. 3).

**There is no universal model.** The first question that we addressed was whether some models systematically outperform others, irrespective of the dataset. This was not the case as model performance varied strongly across thermal performance datasets, with no model consistently reaching high AICc weights throughout (Fig. 3). It is worth noting, however, that certain models (e.g., Gaussian, simplified Briere I, second-order polynomial, Mitchell-Angilletta, Eubank) performed relatively well (with AICc weights greater than 0.2) across many datasets.

**Parameter-rich models do not strongly outperform simpler models as sampling resolution increases.** To understand if complex models

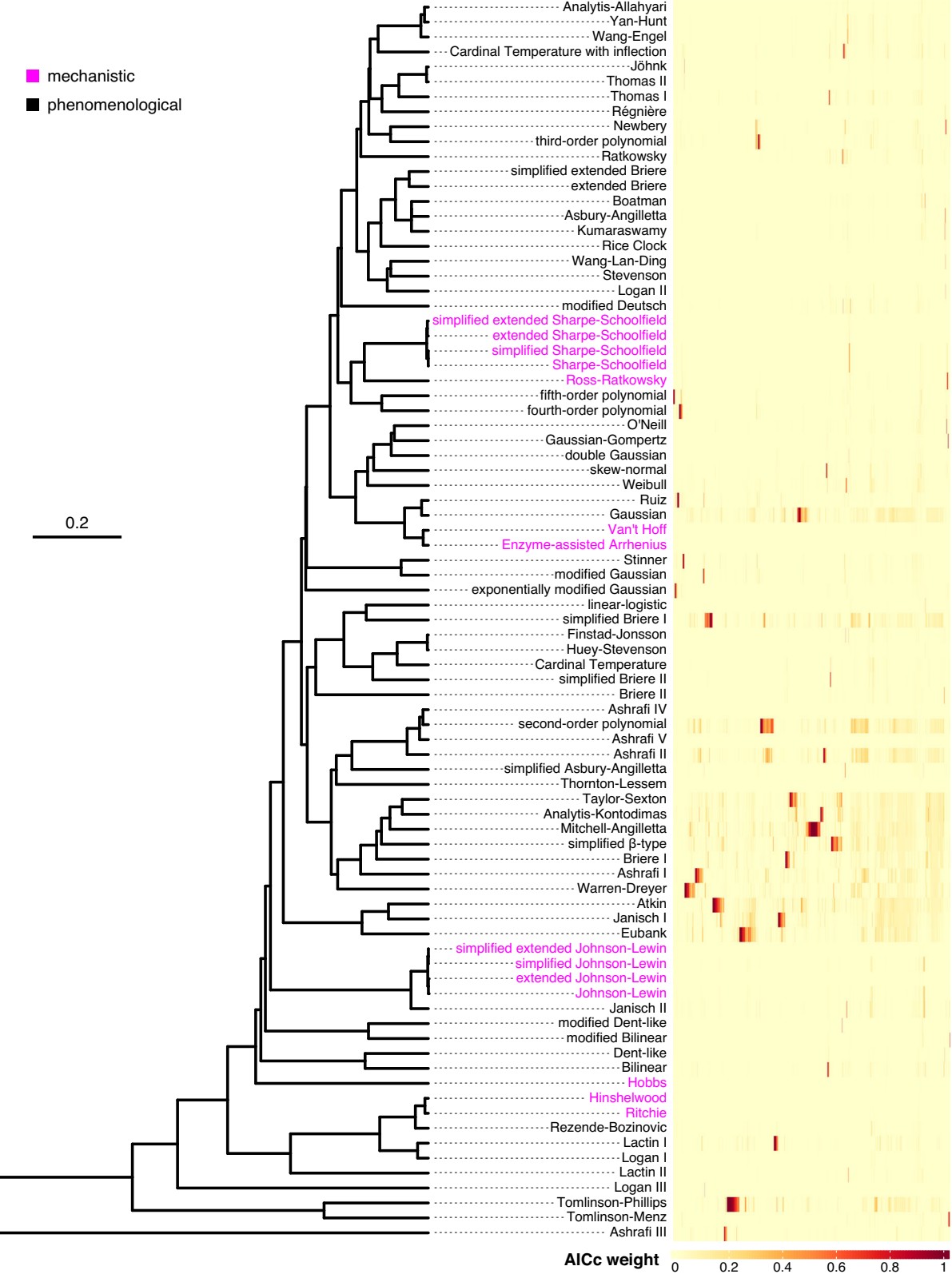

**Fig. 3 | Comparison of the 83 TPC models across 2739 thermal performance datasets.** Left: dendrogram of models based on the similarity of the fits that they produce. Branch lengths represent median Euclidean distances among resulting TPC fits (see "Methods"). Models that branch off very close to each other typically produce nearly identical fits, whereas models separated by large distances tend to produce highly distinct fits. Models based on biochemical kinetics are shown in pink, whereas phenomenological models are in black. Right: The AICc weight of each model (rows) for each dataset (columns). Note that columns have been ordered so that datasets with similar patterns of AICc weights are within close proximity of each other. The figure was rendered using the `ggtree`[80] (v. 2.2.4) and the `ComplexHeatmap`[81] (v.2.12.1) R packages. Source data are provided as a Source Data file.

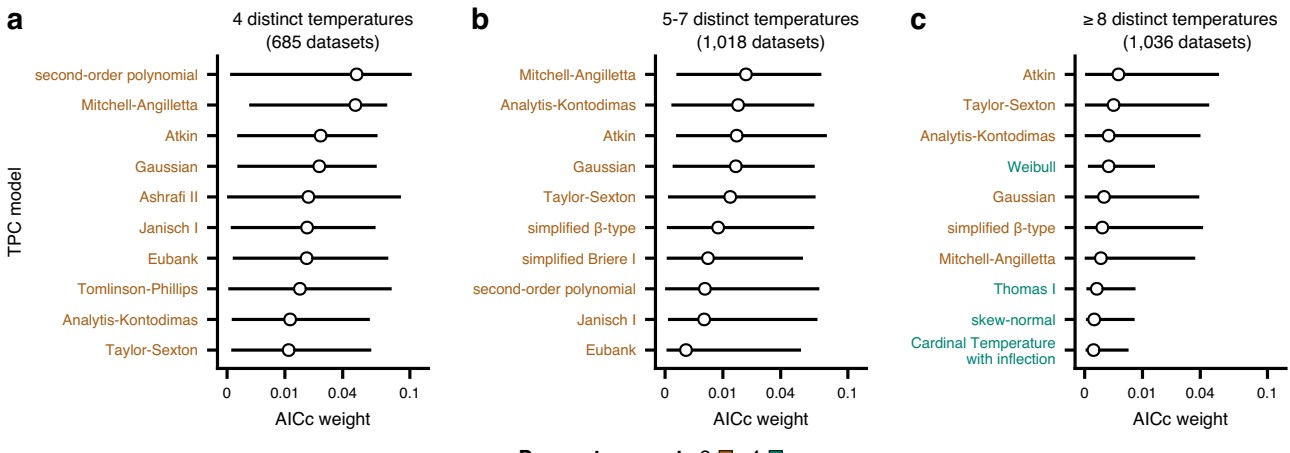

**Fig. 4 | Comparison of model performance across datasets that differ in their sampling resolution.** The ten best-fitting models across datasets of low (**a**; 4 distinct temperatures), moderate (**b**; 5–7 distinct temperatures), or high resolution (**c**; 8 or more distinct temperatures). Circles represent the median AICc weight, whereas horizontal bars stand for the interquartile range. Three- and four-parameter models are shown in brown and green, respectively. Note that values along the horizontal axes do not increase linearly. Source data are provided as a Source Data file.

increasingly outperform simpler alternatives as the number of experimental temperatures increases, we next identified the ten models with the highest median AICc weights across datasets with (a) 4 (low resolution), (b) 5–7 (moderate resolution), or (c) 8 or more distinct temperatures (high resolution). Even for datasets with at least 8 distinct temperatures, the ten models with the highest average performance had either three or four parameters (Fig. 4c). Moreover, datasets of low, moderate, and high resolution shared five three-parameter models among their top ten. It is worth pointing out that the top ten models in the three resolution groups had median AICc weights that were not higher than 0.05 and interquartile ranges that often included values that were effectively zero (Fig. 4).

**Certain phenomenological models consistently outperform mechanistic models, regardless of trait type.** Our third and final question was to determine if mechanistic models (whose parameters describe the activity of underlying rate-limiting enzymes) exhibit better performance than phenomenological models for physiological traits. To this end, we classified each dataset as originating from (a) a physiological, (b) an emergent, or (c) an (organism-organism) interaction trait. Physiological traits directly depend on a small set of biochemical pathways and can be measured as fluxes or rates (e.g., gross photosynthesis rate, aerobic respiration rate). Emergent traits are the ultimate outcome of a wide diversity of cellular biochemical pathways and processes, and are not governed by species interactions. For example, the population growth rate of a sexually reproducing species depends on development rate, fecundity rate, and mortality rate, each of which is, in turn, dependent on a distinct set of biochemical processes that may also vary by life stage. Lastly, traits that explicitly describe species interactions were classified as interaction traits (e.g., resource consumption rate, filtration rate). We then extracted and compared the ten best-performing models across these three groups.

For physiological traits, none of the ten best-fitting models was mechanistic (Fig. 5a), whereas physiological, emergent, and interaction traits shared seven out of ten best-fitting models, albeit with different rankings (Fig. 5a–c, Supplementary Fig. 3 in Supplementary Note 3). We reached similar conclusions after examining the ten best-fitting models for three vastly different traits with large numbers of thermal performance datasets across our data compilation: respiration rate (a physiological trait), population growth rate (an emergent trait), and resource consumption rate (an interaction trait). The three traits had six models in common among their ten best-fitting ones (Fig. 5d–f,

Supplementary Fig. 4 in Supplementary Note 3), whereas the interquartile ranges of AICc weights ranged from ≈ 0 to ~ 0.13. These results show that model performance is not strongly trait-dependent and that even when measurements come from traits that are governed by specific biochemical pathways, mechanistic models do not tend to outperform simpler phenomenological alternatives.

To further explore this unexpected finding, we searched the literature for datasets of the performance of individual enzymes at multiple temperatures. Our search yielded 60 such datasets, 56 of which passed our filtering criteria (see Fig. 2d and the Methods section). Fitting all 83 models to the enzyme datasets[54] revealed a strikingly similar result (Supplementary Fig. 5 in Supplementary Note 4) to that obtained for trait datasets (Fig. 3). More specifically, we detected remarkable variation in model performance across datasets, with mechanistic models not consistently outperforming popular non-mechanistic alternatives such as the Briere I model. This may help explain why the performance of the mechanistic model subset does not vary systematically between physiological, emergent, and interaction traits (Fig. 5).

**Predicting the optimal TPC model**
Finally, we employed machine learning to uncover any rules for selecting the optimal TPC model(s) for a given thermal performance dataset. For this, we compiled 29 variables that describe the data in our study in four main aspects: (i) the type of trait measured, (ii) taxonomic information about the organism (kingdom and phylum), (iii) the shape of the TPC (e.g., its breadth, the degree of symmetry before and after the peak), and iv) information related to sampling resolution (e.g., the number of distinct temperatures before the peak of the curve). The full list of variables, along with their descriptions, is available in Supplementary Table 2 in Supplementary Note 5. We then randomly split the trait data into training and testing subsets (80% and 20% of the data, respectively) and fitted multi-output conditional inference regression trees[59] with all possible combinations of the four aforementioned groups of predictors using the R package `partykit`[60] (v. 1.2-17). This method aims at predicting multiple response variables simultaneously (here, the AICc weights of all models) based on binary splits using the values of predictor variables, selected through non-parametric tests. It ultimately yields data subsets (leaf nodes) that differ considerably in the composition of AICc weights, with the maximum number of such subsets being equal to two raised to the depth of the tree. For practical purposes, we set the maximum tree

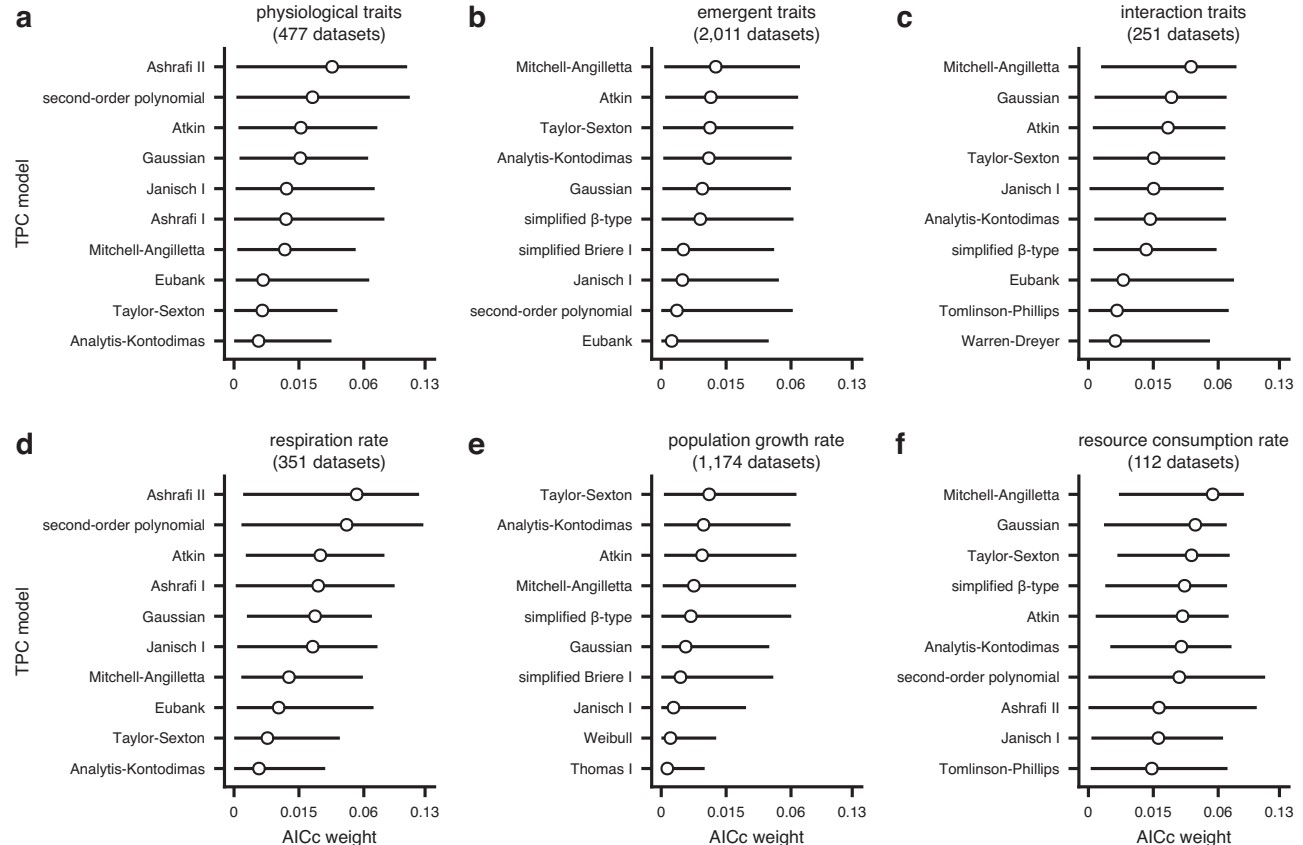

**Fig. 5 | Comparison of model performance across trait groups.** The ten best-fitting models across thermal performance datasets from the three main trait groups (**a**–**c**) and for one representative trait per group (**d**–**f**). Circles represent the median AICc weight, whereas horizontal bars stand for the interquartile range. Note that values along the horizontal axes do not increase linearly. Source data are provided as a Source Data file.

depth to four, allowing the algorithm to generate predictive rules for up to 16 different groups of thermal performance datasets. We also grouped rare levels of categorical predictors (those with fewer than 10 datasets) as 'other' to facilitate training. We determined the best-performing tree based on the $R^2$ value achieved on the training set and also evaluated its performance across the separate testing subset.

Obtaining a tree that could accurately predict the AICc weights for a given dataset proved impossible as the best-fitting tree (Supplementary Fig. 6 in Supplementary Note 6) had $R^2$ values of only 0.12 and 0.1 on the training and testing data subsets, respectively (Fig. 6a, b). More worryingly, the maximum predicted AICc weight by the tree was not greater than 0.22 (Fig. 6b). When we examined how accurately the tree predicted the AICc weights separately for each model, we found that the tree achieved its highest $R^2$ values (below 0.19) for models with a median AICc weight of $\approx 0$ (Fig. 6c). In other words, the tree was slightly better able to predict the models that generally do not represent experimental data well, rather than the models that often provide good fits. Increasing the maximum tree depth to, e.g., 10 (which would raise the number of leaf nodes to 1024) or developing a more advanced predictive method based on deep learning could possibly yield more accurate predictions, but at the cost of limited interpretability.

## Recommendations

All our results converge on the same conclusion, i.e., that there are no simple and well-defined scenarios in which particular TPC models would be consistently favoured over others. This highlights the importance of comparing the performance of multiple alternative models across a given dataset, rather than selecting a single model in

an arbitrary manner. An anticipated objection to this recommendation might be that researchers often choose a particular model because it includes a parameter of interest for their study. We recommend that model selection still be considered because multiple models may share the same parameter. Even when a model does not include the parameter of interest, it may be possible to obtain its value based on other model parameters or from the fitted curve itself. Obtaining estimates of the same parameter using different models additionally allows AICc-weighted model-averaged estimates to be calculated. In cases where none of the fitted models stands out, such estimates should be much more reliable than estimates obtained from a single model. Careful attention should be spent, however, on ensuring that the parameter estimates to be averaged are fully equivalent across models, otherwise the results obtained from averaging will be meaningless. Besides parameter estimates, model averaging can also be applied to the entire TPC[61], enabling a more objective quantification of its shape.

We should clarify that we do not expect (or recommend) that all possible models be fitted to each new thermal performance dataset. Instead, one could examine the dendrogram that we estimated in the present study (Fig. 3) to choose a subset of sufficiently distinct models (i.e., those branching far from each other) for fitting to a dataset. By using our dendrogram as a guide, researchers no longer need to compare the equations of different models manually, which can often be quite challenging in our experience. For example, our dendrogram shows that the Johnson-Lewin model[29] and a common modification thereof (the simplified Johnson-Lewin model; see Supplementary Note 7) yield effectively identical fits, consistent with the conclusions of Yin[62]. It is also worth pointing out that the fits of the mechanistic Johnson-Lewin model (and its variants) are typically highly similar to

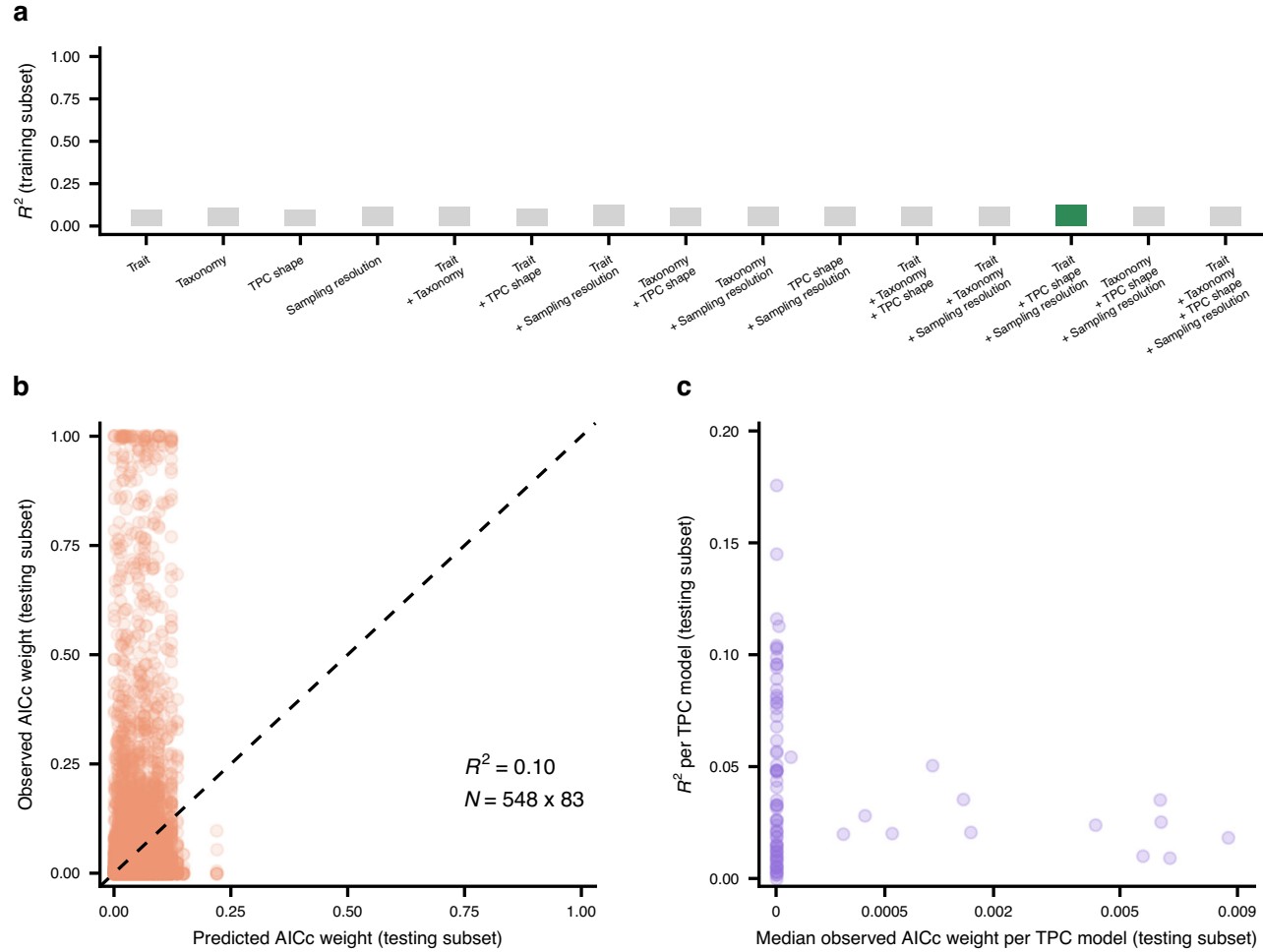

**Fig. 6 | Conditional inference trees with a maximum depth of four cannot accurately predict the AICc weights of TPC models.** Panel (**a**) shows the $R^2$ values reached by alternative candidate trees (with different sets of predictor variables) across the subset of the data on which they were trained. The tree with the highest $R^2$ is highlighted in green. Note that $R^2$ values vary very little across trees (ranging from 0.0946 to 0.1227), with no tree exhibiting sufficiently high performance. Panel (**b**) shows the predictions of the best tree (see Supplementary Fig. 6 in Supplementary Note 6) versus the observed AICc weights across the testing subset (AICc weights of 83 models for 548 thermal performance datasets). The dashed line is the one-to-one line. Panel (**c**) shows the model-specific $R^2$ values achieved by the tree across the testing subset against the median observed AICc weight of each model. Source data are provided as a Source Data file.

the fits of the phenomenological asymmetric Janisch model (Janisch II; Fig. 3), one of the two earliest developed TPC models.

Perhaps the most unexpected result of our study was that phenomenological models generally outperform mechanistic alternatives, even across thermal performance datasets of individual enzymes. These two types of models serve two different purposes and hence occupy different niches in terms of their utility. On the one hand, mechanistic TPC models were developed to estimate key thermodynamic and biochemical kinetic parameters (e.g., enzyme activation energy from the rising part of the TPC; $E$[63]). Estimates of these parameters may be necessary to compare the temperature dependence of processes within and across scales of biological organisation[9,15,38,42,46,64–66] (e.g., whole-organism metabolic rate or community-level fluxes). Conversely, phenomenological models seek to deliver the most statistically reliable characterisation of the shape of a specific TPC, without considering the underlying mechanisms. Because of this, phenomenological models may be better suited for quantifying parameters related to the shape of the TPC, such as critical thermal limits ($T_{min}$ and $T_{max}$). Therefore, we would argue that mechanistic models should not be abandoned due to their (relatively) lower performance, but that their application (or exclusion) should be justified by the aims of a given research study. Moreover, systematic

and strategic model comparison is necessary for theoretical advancement because understanding why certain models perform better than others can accelerate the development of more general theoretical frameworks or the refinement of those existing ones.

No matter which models are selected for fitting to a specific dataset, it is essential that the data allow sufficient statistical power to yield reliable estimates. In particular, in the present study, to avoid overfitting, we only fitted a model to a given dataset if its parameter count was lower than the number of distinct temperatures. In addition to this, researchers should ensure that the TPC is sampled sufficiently in temperature ranges relevant to their parameter(s) of interest. For example, if the goal is to estimate the thermal optimum, there should be numerous data points before and after the peak of the TPC. If this is not the case, the parameter estimates may be imprecise and biased to various degrees[43]. The number of experimental replicates per temperature is another matter that merits consideration, especially for traits for which precise measurements are difficult to obtain (e.g., behavioural traits). The above points do not constitute an exhaustive list but represent some common guidelines that should be relevant to most thermal biology studies.

In conclusion, our study shows that for describing the shape of thermal performance curves, there is no model "to rule them all". This

makes it imperative that researchers adopt multi-model selection approaches to identify the most appropriate one(s) for a given class (e.g., trait type) of data and research question. Such a strategy is likely to yield multiple benefits, such as bridging thermal biology "silos" (e.g., plant physiologists versus environmental microbiologists), improving the rigour and reproducibility of the study, and advancing theory. Ultimately, this will accelerate progress towards achieving a common goal across biological fields: identifying general mathematical models rooted in mechanistic, biophysical principles that can predict the effects of environmental temperature on complex biological systems.

## Methods

### Sources of thermal performance datasets
Our data compilation includes thermal performance datasets of (i) population growth rate of bacteria, archaea, and phytoplankton[66,67], (ii) photosynthesis and respiration rate of algae, aquatic, and terrestrial plants[65], (iii) metabolic rate of zooplankton[64,68,69] and other heterotrophic ectotherms[70], (iv) development rate of damselflies[71] and helminths[72], (v) overall dynamic body acceleration of elasmobranchs[73], and (vi) numerous other biological traits included in the Biotraits database[74].

### Compilation of a TPC model catalogue
To compile our catalogue of TPC models, we queried Google Scholar using terms such as "thermal performance curve model", "thermal performance curve equation", "thermal reaction norm model", "temperature response curve model", etc. From the resulting hits, we progressively added models that were not mathematically equivalent to any model that was already on the list. For example, the equation of the Johnson-Lewin model[29,46] is equivalent to that of a particular 4-parameter variant of the Sharpe-Schoolfield model[44,63].

### Fitting TPC models to thermal performance datasets
Given that many of the models in our catalogue could not support non-positive trait measurements (e.g., negative population growth rate), we first removed such values from the datasets. To reduce the possibility of overfitting, we only fitted a model to a thermal performance dataset if the parameters of the model were fewer than the number of distinct experimental temperatures. For fitting the models separately to each dataset, we used the R package `nls.multstart`[61,75] (v. 1.2.0). To find the optimal combination of parameter values that maximises the fit of a given TPC model to a given dataset, `nls.multstart` repeatedly fits the model using the Levenberg-Marquardt algorithm from different starting values. We specified seemingly realistic starting values for each model in various ways, depending on its parameters. For example, for models that include the $T_{min}$ and $T_{max}$ parameters (see Fig. 1), we set their starting values to the minimum and maximum experimental temperatures in the dataset, respectively. In some cases, we specified starting values by performing linear regression on a subset of the dataset (e.g., to estimate the slope of the rising part of the TPC) or by using parameter estimates from the literature. We then fitted each model to each thermal performance dataset 1000 times. In every iteration, we used a different starting value per parameter, sampling from a uniform distribution with bounds equal to 0.5 and 1.5 times our initial starting value. To further facilitate the overall process, we fitted models to trait measurements on the natural logarithm scale, increased the stringency of the convergence criteria by modifying the Levenberg-Marquardt parameters `ftol`, `ptol`, `maxiter`, and `maxfev`, set appropriate bounds for each TPC parameter where possible, and applied other checks (e.g., we forced $T_{pk}$ to be $>T_{min}$ and $<T_{max}$). The source code for fitting the models is available from GitHub and also archived on Zenodo[76].

We calculated the AICc values of all model fits with an $R^2$ of at least 0.5. In a limited number of thermal performance datasets, the AICc value of at least one model was equal to $-\infty$, which is indicative of severe overfitting. When this was the case, we progressively removed the most parameter-rich models and examined the AICc values for that dataset again. If we still obtained values of $-\infty$ even when only three-parameter models were included, we excluded the corresponding dataset from our analyses. After this process, we ended up with 2739 datasets out of the initial 3598. Next, we constructed a table of AICc weights per model and per dataset. If a model could not converge on a specific dataset or was excluded for any of the aforementioned reasons, we assigned an AICc weight of 0.

### Inferring a dendrogram of TPC models
To systematically compare the models in our catalogue, we first generated the curve for each model fit with an $R^2$ value greater than or equal to 0.5, across the 2739 thermal performance datasets. This was done from the minimum experimental temperature of each dataset to the maximum temperature, with a step size of 0.1 °C. We then constructed a matrix of pairwise Euclidean distances among curves (and, hence, models) per dataset, based on all aforementioned temperatures. Given that datasets varied considerably in the magnitude of their trait values, we performed maximum normalisation[77] (also known as "l-infinity normalisation") to ensure that pairwise distances were comparable across datasets. More specifically, we corrected each dataset-specific distance matrix by dividing its values by the maximum height of all corresponding curves. We summarised the resulting distance matrices into a single (average) matrix by taking the median of each cell across all matrices where possible. Finally, based on the average matrix, we calculated a dendrogram of TPC models using hierarchical clustering with average linkage, i.e., with the UPGMA[78] algorithm, which has also been used for distance-based phylogenetic tree inference. Models that were placed close to each other in the dendrogram would generally produce very similar fits.

### Reporting summary
Further information on research design is available in the Nature Portfolio Reporting Summary linked to this article.

## Data availability
The thermal performance datasets included in this study are available from Figshare[54]: https://doi.org/10.6084/m9.figshare.24106161.v3. These datasets were collected from numerous previous studies[64,66–73] or from the Biotraits database[74]. Source data for all figures are provided in this paper as a Source Data File.

## Code availability
The source code for the main analyses of this study[76] is available at https://github.com/dgkontopoulos/Kontopoulos_et_al_83_TPC_models_2024 and archived at https://doi.org/10.5281/zenodo.12608191.

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

## Acknowledgements

A.S. was supported by the ANR project EcoTeBo (ANR-19-CE02-0001-01) from the French National Research Agency (ANR). S.P. was funded by the Leverhulme Research Fellowship RF-2020-653\2 and UK national NERC grants NE/M020843/1 and NE/S000348/1. We are grateful to Alejandro Isla, Alexander Kazhdan, Perng Lee, Olivia Morris, Richard Sheppard, and Joss Thomas for providing some of the thermal performance datasets used in this study.

## Author contributions

Conceptualisation: D.G.K., A.S., A.I.D., and S.P. Data curation: D.G.K. Formal analysis: D.G.K. Methodology: D.G.K., A.S., A.I.D., and S.P. Project administration: D.G.K. Resources: A.S., M.D., N.G., A.I.D., and S.P. Software: D.G.K. Visualisation: D.G.K. Writing – original draft: D.G.K. Writing – review & editing: all authors.

## Competing interests

The authors declare no competing interests.
