## [Peer Review File · Nature Communications]

No universal mathematical model for thermal performance curves across traits and taxonomic groups

Corresponding Author: Dr Dimitrios Kontopoulos

Version 0:

Reviewer comments:

Reviewer #1

(Remarks to the Author)

Kontopoulos and colleagues compiled data on >1000 thermal performance curves from the literature to provide guidance on TPC model selection. I found the paper very well-written, and the methods easy to follow. Their conclusion was not particularly surprising, but important nonetheless given its statistical rigor. Their recommendations were well-founded, and I had only a few minor comments to improve clarity (see below).

L36: There is also evidence that physiological processes in endotherms also exhibit TPCs (e.g., see Butler et al. 2013, Am Nat).

L59: As written, this indicates that TPCs for physiological traits are constrained by single-rate limiting enzymes. It is more likely that they are limited by a more than one enzyme across a range of temperatures. I suggest re-wording this sentence to "activity of a limited number of rate-limiting enzymes".

L66: This is a good point that the authors may want to address if the goal is to develop a 'best practices' for fitting TPCs. That is, should there be a minimum number of experimental temperatures? Based on Fig. 4, it looks as if experiments using fewer experimental temperatures yield more predictive, better fitting TPCs?

L109: Were performance metrics (dependent variables) all measured at constant temperatures? More recent investigations have begun to examine the role of temperature fluctuation (e.g., 20+/-5C vs. 25+/-5C vs. 30+/-5C) and its influence on TPCs (e.g., Kingsolver 2015, J Exp Biol). Fluctuating temperatures are more ecologically relevant and can yield different results than constant temperature-derived TPCs.

L172-173: Does this mean symmetry, in particular, or were other characteristics of TPC shape considered here (e.g., breadth vs. peakiness)?

Reviewer #2

(Remarks to the Author)

Kontopoulos and colleagues have completed an admirable amount of work to compare the performance of 83 thermal performance models across 2,739 trait datasets. While their investigations do not reveal a consistent top-performer, they provide some useful insights to the research community.

I have inserted questions into the PDF of the manuscript and supplementary materials but make a few overarching comments here.

1. Study novelty/contribution. The authors indicate that previous studies were limited in their scope and that no other study to date has had the coverage in terms of number and types of models, types of traits and diversity of taxonomic groups. Quantitative comparisons would help show how comprehensive the present study is.

2. Model representativeness. A google scholar search was used to find TPC models but I observed that a few models

applied in seminal papers were not included. One such model is Norberg et al. 2004 L+O 49(4, part 2), 2004, 1269–1277 [cited in Thomas et al. 2012 A global pattern of thermal adaptation in marine phytoplankton (722 cites)]. Given that highly cited works are influential in the field, it is worth including this model in the comparison, or at the very least indicating which tested model it is similar to. A note was made about models that were excluded – if models such as this did not meet the criteria (of being able to fit one trait dataset at a time), then these should be made explicit.

3. Model performance. I struggled with the term AICc weight. A low AIC is desirable but what is AIC weight and is low weight better than high weight? Line 132 – what does perform 'relatively well' mean? Describe quantitatively.

4. Empirical data. Indicate what low, medium and high resolution is in the text as well as legend of Fig. 4. Also, for the descriptors of empirical datasets, these should be integrated as subheadings into the supplementary table. [Do they relate to row colours?]

5. Guidance for model use. I am surprised that the mechanistic models did not perform better than the phenomenological ones – what are the implications for physiologists? Are they wasting their time? Is there any advice to offer about increasing replication at each temperature, increasing temperature resolution or range? The machine learning aimed to test the factors in the empirical data that may affect model fit, but the result was not very satisfying, with no factors being identified as more influential than any other. Here we are left with the knowledge that not all models perform equally, with the recommendation that users consider several thermal performance models from different clusters in the tree. Can the conclusions consider a broader set of use cases for how TPCs are being applied in thermal niche models, what considerations ecosystem modelers might have (amongst multiple trophic levels) etc? The authors mention population, ecosystem and disease transmission dynamics in the abstract but do not elaborate on these scenarios in the conclusions and recommendations. Some general guidelines on decisions for choosing models that represent more complex use cases would leave the reader more informed.

6. Manuscript structure. I appreciated why the authors chose to combine the results and discussion however the methods were also incorporated into this section. Results were also found in figure legends. This meant that the Materials and Methods were rather thin and did not contain information I expected (or left me asking where it was). It read as if the Methods was written after the rest of the manuscript to deal with any omissions. Some relocation of the text or improved explanation in the methods is recommended to more easily reproduce the current study.

Reviewer #3

(Remarks to the Author)

This work aims to guide model selection for thermal performance curves, applying 83 different models to a collation of relevant data sets. The key result is as stated in the title: no model is optimal for the scenarios tested (different numbers of temperatures or different traits), though I also found it interesting that mechanistic models do not tend to outperform simpler phenomenological alternatives, and did not gain much in the way of insights about different taxonomic groups

I think the paper is clearly written and generally achieves what it sets out to do, with a number of caveats. The main ones to me are as follows.

i) I'm not quite convinced of the significance of the work: I'm unaware of anything similar, and don't believe the authors overstate its importance. But I do wonder how broadly appealing it will be, given many of us fit TPCs for biological insights (e.g., comparisons of different species or contexts) without the exact model being a primary concern. I also wondered if there was a missed opportunity to identify other gaps in the kind of thermal performance data we're sampling – for example, do we only focus on adults, or Gaussian traits? Are we sampling sufficient temperatures to fit curves well (unclear from Fig 4)? With so many candidate models but narrow frame of reference biologically, the recommendations and conclusions are potentially limited in scope.

ii) The outcomes of the conditional inference tree analysis are worrying, and aren't really delved into. What are the possible reasons for such a poor result? Some explanation, probing, or justification for the striking patterns in Fig 6 seem warranted to put this result in better context.

iii) On a similar note, I often found myself wanting more explanation and justification for decisions taken throughout the comparisons. I list these against specific line numbers below.

L39. Yes, TPCs are usually asymmetric, though Angilletta also notes that curves for survivorship during embryonic or larval stages tend to be more symmetrical. This is acknowledged for per capita fecundity on L77, but are such instances exceptions to the rule, or does the rule emerge from limited sampling of certain traits? I guess I'm querying if this point is worth emphasising earlier.

L62-63. I'm curious: is number of temperatures the sole source of high resolution, or could resolution be increased by other aspects of experimental design? For example, number of independent replicates per temperature, or clustering of temperatures around thermal optima (Tpk) or limits?

This led me to wonder if any insights can be gained about *how* models perform better or worse. I can think of instances where thermal optima appear to match data well, but thermal limits do not – for example, because performance was not sampled sufficiently at extremes. If researchers are interested in a certain TPC feature, should they choose one model over

another to estimate that feature well?

L92. Does aim 3 explicitly compare mechanistic versus phenomenological models with the same numbers of parameters? Is this important for fair comparison, or moot because model types tend to have similar complexity? Please clarify so readers can follow the logic.

L104 and Fig 2. All Gaussian traits? At what life stages? Are there other important dimensions worth acknowledging, given this is in some ways as systematic review with value for highlighting knowledge gaps? Some of these issues are gaining attention, so it feels like an oversight worth addressing.

L122-124. Readers are told what was done but not why. Please clarify why an R2 value of 0.5 was chosen as the relevant cut-off.

L124. What is AICc weight? The per-model proportion of total AICc values? Please clarify for readers.

L129-135. This is the message of the paper flagged in the title. I think readers could be forgiven for wanting some biological context, rather than a list of models that performed relatively well. I realise the stated aim of the work is being delivered on, but wonder if the scope is somewhat narrow for the intended audience.

L135. Why focus on the top ten models in each of the scenarios in Figs 4 and 5? Are results robust to this decision? Please clarify so readers can follow the logic.

L130. Perhaps I missed it, but what should readers interpret as high AICc weights, or weights that support adequate model performance?

Fig 3 legend. 'Branch lengths represent median Euclidean distances among resulting TPC fits': it would be worth clarifying for readers how they should interpret the branch lengths of models versus the arrangement (or clustering) of models in the dendrogram.

L136. Do i), ii), and iii) correspond to (A), (B), and (C) in Fig 4? It would be worth using the same labelling if so.

L139-142. Again, it would help readers to know exactly what AICc weights represent, or what weights support adequate model performance. They're mainly used to rank models, yet the point here (that the best models in all three groups had median weights of ≤ 0.05 and interquartile ranges including weights of effectively zero) seems to encourage interpretation on the absolute scale. What should we infer (here and in Fig 5) from medians being so small, when some models have weights near 1? That models tend to have little or no predictive value, or that most models are *relatively* bad and swamp high-weight models when presented this way? Likewise, what should we infer from lower medians for models fitted to higher resolution data than for models fitted to lower resolution data (or different trait groups in Fig 5)? Apologies if I'm misunderstanding, but if I was confused I suspect other interested readers will be too.

L188-198. Indeed, this and patterns in Fig 6 are worrying. Was the problem explored to diagnose what could be driving it? For example, do I understand correctly that models were fitted to 29 predictors (I wasn't sure what fitting to 'different combinations of predictors' meant)? Did simpler models, models with uncorrelated predictors, or models with fewer response variables, give better prediction?

L211-222. I guess these seem like limited take-homes for this work. Why focus on mechanistic models here if they perform no better than phenomenological ones? Fitting truly nonlinear models is challenging -- even more so for non-Gaussian traits, or experimental designs that warrant mixed model approaches. Why not opt for phenomenological models, such as polynomials that can accommodate these scenarios and still allow derivation of important features (Tmin, Tmax and Tpk) in the manner of, for example, Kellermann et al 2019?

Version 1:

Reviewer comments:

Reviewer #1

(Remarks to the Author)

I appreciate the authors' efforts to incorporate comments from reviewers. The revised version of the manuscript is much-improved and (to me) ready for acceptance.

Reviewer #2

(Remarks to the Author)

Kontopoulos and colleagues have revised their manuscript in view of reviewer comments and have responded with authenticity to the feedback.

Inclusion of enzyme TPC data added depth to the analysis, and I thought the Box was a particularly effective solution to explaining the differences in model performance metrics.

I have only a few small comments remaining.

1. Line 65: 'parameter-rich models will systematically outperform simple alternatives, e.g., revealing two different gradients at the rising part of the TPC'. Could the models not also have divergent performance at the descending part of the TPC?
2. Line 101: What did the filtering process exclude? Why were >800 datasets excluded from the analysis?
3. Line 298: Why not negative instead of non-positive?

Reviewer #3

(Remarks to the Author)

I think the authors have done a good job in revising this manuscript, and find it much improved from the version I reviewed originally. The main results remain unchanged, but the explanations of them have been made clearer, as has their broader relevance.

In particular, the addition of Box 1 (explaining the model performance metrics used) goes a long way to resolving my comments about the meaning and application of these metrics.

I especially appreciate the substantial expansion of the Recommendations and Conclusions section, which to me makes the outcomes of the work more broadly useful, in terms of developing best practices for fitting TPCs to data.

I found the authors' justifications in cases where few/no changes were made to be reasonable.

I had only one *very* minor point: should 'whereas' in L175 be 'whereby'? Perhaps I misunderstand, but 'whereas' seems to convey the opposite meaning to that intended.

Responses to the Reviewers

Comments made by the Reviewers are in **green**, whereas our responses are in **black**.

Comments by Reviewer #1

1. Kontopoulos and colleagues compiled data on >1000 thermal performance curves from the literature to provide guidance on TPC model selection. I found the paper very well-written, and the methods easy to follow. Their conclusion was not particularly surprising, but important nonetheless given its statistical rigor. Their recommendations were well-founded, and I had only a few minor comments to improve clarity (see below).

L36: There is also evidence that physiological processes in endotherms also exhibit TPCs (e.g., see Butler et al. 2013, Am Nat).

We thank the reviewer for their thoughtful assessment. We were in fact surprised that mechanistic, thermodynamically-motivated models did not perform systematically better when it came to traits at lower levels of organization.

As far as endotherms are concerned, we agree, but given that the explicit focus of our paper is on ectotherms, we feel that mentioning endotherms would distract from the main narrative / focus.

2. L59: As written, this indicates that TPCs for physiological traits are constrained by single-rate limiting enzymes. It is more likely that they are limited by a more than one enzyme across a range of temperatures. I suggest re-wording this sentence to “activity of a limited number of rate-limiting enzymes”.

We agree with the Reviewer that trait performance across a range of temperatures is more likely to be governed by multiple, rather than a single rate-limiting enzyme. Nevertheless, in line 59 we wanted to point out that many “mechanistic” mathematical models were developed under the assumption of a single rate-limiting enzyme. We now explicitly mention that this assumption may be unrealistic in lines 62–63.

3. L66: This is a good point that the authors may want to address if the goal is to develop a ‘best practices’ for fitting TPCs. That is, should there be a minimum number of experimental temperatures? Based on Fig. 4, it looks as if experiments using fewer experimental temperatures yield more predictive, better fitting TPCs?

We apologise for not clearly defining AICc weights in the manuscript which led to some confusion, as is evident from all three Reviewers’ comments. The AICc weight is not a measure of **absolute** fit of a model to the data (this is captured by the R^2 ; see Supplementary Fig. 1), but rather a measure of how well a model fits the data **relatively to all other fitted models**. Thus, the negative relationship between AICc weights and sampling resolution in Fig. 4 mainly reflects that a) larger sample sizes enabled us to fit more models and b) no model consistently outperformed others. In a scenario where a single model would systematically achieve much better performance than its alternatives, this model would have an AICc weight close to 1, whereas all other models would have AICc weights close to 0.

We now explicitly define AICc weights and their purpose in Box 1.

4. L109: Were performance metrics (dependent variables) all measured at constant temperatures? More recent investigations have begun to examine the role of temperature fluctuation (e.g., 20+/-5C vs. 25+/-5C vs. 30+/-5C) and its influence on TPCs (e.g., Kingsolver 2015, J Exp Biol). Fluctuating temperatures are more ecologically relevant and can yield different results than constant temperature-derived TPCs.

We agree with the Reviewer on the greater biological realism of fluctuating temperatures. It

has prompted us to clarify the potential importance of acclimation time in TPC experiments in assessing model performance. Almost all thermal performance datasets analysed in this study were obtained under fixed temperatures. Thus, even before we consider fluctuating temperatures, one issue that needs to be addressed is whether the timescale of acclimation to each fixed temperature in a TPC changes which model is most appropriate [1]. We have now added a few sentences (lines 103–111) where we explicitly address this.

5. **L172-173: Does this mean symmetry, in particular, or were other characteristics of TPC shape considered here (e.g., breadth vs. peakiness)?**

Some of our 29 descriptor variables provided information a) on the symmetry of the TPC before and after the thermal optimum, b) on the TPC breadth, and c) on the ratio of the minimum to maximum trait measurements (which captures the degree of “peakiness” in the data). These are all described in Supplementary Section S5 and we have now made this clearer in lines 185–187 of the main text.

Comments by Reviewer #2

6. **Kontopoulos and colleagues have completed an admirable amount of work to compare the performance of 83 thermal performance models across 2,739 trait datasets. While their investigations do not reveal a consistent top-performer, they provide some useful insights to the research community.**

I have inserted questions into the PDF of the manuscript and supplementary materials but make a few overarching comments here.

Study novelty/contribution. The authors indicate that previous studies were limited in their scope and that no other study to date has had the coverage in terms of number and types of models, types of traits and diversity of taxonomic groups. Quantitative comparisons would help show how comprehensive the present study is.

Thank you for your kind words and for your thorough and constructive comments. We now describe in Supplementary Table 1 the taxonomic breadth, traits, model count, and number of thermal performance datasets of previous studies.

7. **Model representativeness.** A google scholar search was used to find TPC models but I observed that a few models applied in seminal papers were not included. One such model is Norberg et al. 2004 L+O 49(4, part 2), 2004, 1269–1277 [cited in Thomas et al. 2012 A global pattern of thermal adaptation in marine phytoplankton (722 cites)]. Given that highly cited works are influential in the field, it is worth including this model in the comparison, or at the very least indicating which tested model it is similar to. A note was made about models that were excluded – if models such as this did not meet the criteria (of being able to fit one trait dataset at a time), then these should be made explicit.

The Norberg, 2004 model is mathematically equivalent to the one used by Thomas et al., 2012 and is called “Thomas I” in our list. This is because we first came across the Thomas et al., 2012 study when searching the literature for models. We now note in the Supplementary Information that the “Thomas I” model is equivalent to that fitted by Norberg, 2004.

Regarding models that were not able to fit a single thermal performance dataset at a time, we only came across one such model (by Corkrey et al., 2012), which we cite in line 116.

8. **Model performance.** I struggled with the term AICc weight. A low AIC is desirable but what is AIC weight and is low weight better than high weight? Line 132 – what does perform ‘relatively well’ mean? Describe quantitatively.

Regarding AICc weights, see our response to comment #3 above and Box 1. We have also added the words “(with AICc weights greater than 0.2)” in line 133 to clarify what we mean by

“relatively well”.

9. Empirical data. Indicate what low, medium and high resolution is in the text as well as legend of Fig. 4. Also, for the descriptors of empirical datasets, these should be integrated as subheadings into the supplementary table. [Do they relate to row colours?]

We have now made those more explicit in lines 136–137, in the caption of Fig. 4, and in the caption of Supplementary Table 2.

10. Guidance for model use. I am surprised that the mechanistic models did not perform better than the phenomenological ones – what are the implications for physiologists? Are they wasting their time? Is there any advice to offer about increasing replication at each temperature, increasing temperature resolution or range? The machine learning aimed to test the factors in the empirical data that may affect model fit, but the result was not very satisfying, with no factors being identified as more influential than any other. Here we are left with the knowledge that not all models perform equally, with the recommendation that users consider several thermal performance models from different clusters in the tree. Can the conclusions consider a broader set of use cases for how TPCs are being applied in thermal niche models, what considerations ecosystem modelers might have (amongst multiple trophic levels) etc? The authors mention population, ecosystem and disease transmission dynamics in the abstract but do not elaborate on these scenarios in the conclusions and recommendations. Some general guidelines on decisions for choosing models that represent more complex use cases would leave the reader more informed.

We were similarly surprised by the lack of clear winners among models when considering specific subsets of the data. For example, we thought that mechanistic models that explicitly include enzyme/biochemical kinetics would outperform others when fitted to data on individual-level metabolic rates (primarily, respiration and photosynthesis) because these traits are closest to the actual processes being modelled as well as satisfying the underlying assumptions. On the contrary, more phenomenological models would potentially perform relatively better when it came to more “emergent” traits (e.g., population growth rate, development rate), which stem from a number of complex metabolic processes.

We have now collected 56 additional datasets with measurements of the performance of individual enzymes at multiple temperatures. After fitting all 83 models to each new dataset, we found that phenomenological models generally outperform their mechanistic alternatives, similarly to physiological trait datasets. These are described in lines 170–179 of the main text, as well as in Supplementary Fig. 5.

In our view, all these results suggest that mechanistic and phenomenological models are appropriate for different purposes. The purpose of phenomenological models is to statistically describe the observed patterns as best as possible, without being constrained by mechanistic assumptions. Such models are often incorporated in non-mechanistic species distribution models. In contrast, the primary purpose of mechanistic models is not to provide the most accurate statistical representation of the data, but to estimate key thermodynamic and biochemical parameters (e.g., the activation energy; E) that cannot be obtained from phenomenological models. Such parameters can provide insights into the processes that underlie the temperature dependence of a given trait and their links to processes at higher levels of biological organization.

We should also note that when a particular mechanistic model fits a dataset very well, its predictions beyond the range of the data are expected to be usually more accurate than those of a phenomenological model. This is because the former is constrained by the underlying mechanisms which, in this hypothetical case, provide an accurate representation of the observed pattern.

We have now added a detailed discussion of the differences between mechanistic and phenomenological models, and our recommendations for their application in lines 242–259.

11. Manuscript structure. I appreciated why the authors chose to combine the results and discussion however the methods were also incorporated into this section. Results were also found in figure legends. This meant that the Materials and Methods were rather thin and did not contain information I expected (or left me asking where it was). It read as if the Methods was written after the rest of the manuscript to deal with any omissions. Some relocation of the text or improved explanation in the methods is recommended to more easily reproduce the current study.

We have now moved some methodological details from the “Results and discussion” section to the “Methods” section. Any methodological information that still remains in the “Results and discussion” section is necessary for the interpretation of the results.

Comments by Reviewer #2 added on the PDF files of the main text and the Supporting Information

12. Line 19 - “ecological”: later in the ms I would like to see examples of traits.

Some representative examples of traits included in this study are already provided in Figs. 2 and 5, in the subsection “Sources of thermal performance datasets”, and elsewhere.

13. Fig. 1B: none of the models look asymmetric like the LHS example

In fact, only the Janisch I fit is symmetric before and after the thermal optimum, as mentioned in the figure caption. To make this clearer, we have now added dashed lines at the thermal optimum estimate of each model fit.

14. Fig. 1 caption - “*Erigeron annuus*”: please indicate what this species is - a fish or???

It is a flowering plant and its common name is “eastern daisy fleabane”. We have now added this to the caption of Fig. 1.

15. Lines 68–69: provide some typical numbers so that the coverage and comprehensiveness of this study can be compared

See our response to comment #6 above.

16. Line 88: ie are there models that are more universal than others?

We believe that the word “universal” is more ambiguous than the phrasing that we have used (“consistently outperform others across any dataset”).

17. Lines 89–90: define parameter-rich

And why ask this question only of high resolution datasets? it’s not immediately clear

We have now rephrased this sentence to avoid confusion: “Do models with more parameters tend to outperform simpler alternatives as the sampling resolution of the dataset increases?”

18. Lines 91–92: in the methods i’m looking for physiological mechanisms to be explained

We already provide a justification for this question earlier in the Introduction (lines 53–61) and in the Results and discussion section (lines 144–156).

19. Line 101: is the data compilation found anywhere - eg supplementary?

We had deposited it to Figshare; see the “Data availability” section.

20. Fig. 2: Regarding the datasets - how many were high resolution? define high resolution vs medium or low

We define resolution in terms of the number of distinct temperatures later, in lines 136–137. As we describe in lines 261–263 and 322–327, we ensured that a) there was at least 1 distinct

temperature more than the number of parameters of each model that we fitted, and b) there were no models with $AICc = -\infty$.

21. Lines 114–115: did this exclude models used in seminal papers? it may be worth citing them to show readers that they were considered but didn't meet this study's criteria

See our response to comment #7 above.

22. Line 133: what is relatively well? quantify

See our response to comment #8 above.

23. Line 133: this analysis needs to go another step. WHY? what is it about the models or the datasets that leads to this?

Our expectation was that model performance would be linked to sampling effort or trait information, but this was not supported by our results. From a statistical standpoint, these models have few parameters and simpler mathematical formulations, which can make them easier to fit compared to other alternatives. See also our response to comment #10 above.

We have now added a figure in the Supplementary Information (Supplementary Fig. 2) that shows the number of thermal performance datasets that each model was able to fit. Based on the figure, the top models according to the number of fitted datasets are the simplest 3-parameter ones (e.g., Mitchell-Angilletta, Atkin, Eubank, Gaussian). Nevertheless, the figure also shows that no single model could reliably fit every single one of our thermal performance datasets.

24. Line 137: define resolution

We have now rephrased this sentence as “datasets with at least 8 distinct temperatures”.

25. Line 140: state in words what AICc weight is

See our response to comment #3 above.

26. Fig. 4 - “Model performance ... (green).”: not text for the legend

We have now removed these two sentences from the caption of Fig. 4.

27. Fig. 5 - a lower AIC score is better, but it's not clear whether a high AICc weight is better. is the model closest to the plot origin on the y-axis the better performing model? it's a bit confusing

See our response to comment #3 above.

28. Fig. 5 - “Similarly to ... of the same trait.”: not appropriate for legend

We have now removed these two sentences from the caption of Fig. 5.

29. Line 172: supplementary list should be organised into broad headings that reflect those mentioned in the text

We now explicitly mention the four groups of TPC descriptors in the caption of Supplementary Table 2.

30. Line 173: isn't this described elsewhere as resolution?

We have now rephrased this sentence to avoid confusion.

31. Line 175: RANDOMLY?

Randomly, indeed. We have now added this word in line 188.

32. Line 188: is accurate the right word to use here? In my mind the tree represents the factors that contribute to the overall level of fit (AICc). From what I understand happens in the algorithm, it had difficulty converging on a tree with the number of nodes specified.

We have now rephrased that sentence in line 201 to avoid confusion.

33. Fig. 6 - change the y-axis scale so that differences in the x-axis categories are easier to see

The values along the vertical axis range from 0.0946 to 0.1227. Given this very limited range of values, we believe that it is more meaningful for the vertical axis to range from 0 to 1, as this makes it clear that no combination of predictors achieved a high accuracy. We now state this clearly in the text legend of the figure.

34. Lines 225–229: Choose the most appropriate model - ie with the best fit. the conclusion doesn't address the comparability aspect amongst different taxa or across different levels of biological organisation. What ELSE does this exercise yield in terms of advice? What are the end-user archetypes and the questions they are asking?

Please see our response to comment #10 above. Briefly, the main conclusions of our study are:

- (a) There are no well-defined scenarios for choosing one or more models for a given type of trait or taxon.
- (b) Certain phenomenological models will generally outperform mechanistic alternatives. Thus, the choice between mechanistic and phenomenological models should be driven by the goals of a given study.
- (c) Researchers can take advantage of our dendrogram of models (Fig. 3) to select a diverse set of models to fit to their datasets.

We have now also added a paragraph with further recommendations regarding obtaining data with sufficient statistical power for addressing particular research questions (lines 260–271).

35. Line 251: shown somewhere?

Our 83 models varied a lot in their parameters and, therefore, parameter bounds had to be adjusted to each different model. Information on these bounds can be found in the source code deposited on GitHub (see the Code availability section).

36. Line 277: how many gaps were there?

We have now added a new figure (Supplementary Fig. 2) to show the number of fits per model in our study.

37. Supplementary Fig. 1: units for density?

The units of the vertical axis (probability density) of a density plot are the multiplicative inverse of the units of the variable shown in the horizontal axis. In this case, they would be 1 divided by the R^2 units. Given that R^2 is dimensionless, the vertical axis is also dimensionless.

38. Supplementary Fig. 2: would be useful to cross-reference/list the 7 models that were amongst the best performers for physio, emergent and interaction traits.

Supplementary Fig. 3: same comment as above

Addressed as suggested for both Supplementary Figs.

39. Supplementary section S3: some broader level descriptors were included in the text - would be useful if these were listed as subheadings in the table; presumably this is what the different colours are? Explain in the legend

See our response to comment #29 above.

Comments by Reviewer #3

40. This work aims to guide model selection for thermal performance curves, applying 83 different models to a collation of relevant data sets. The key result is as stated in the title: no

model is optimal for the scenarios tested (different numbers of temperatures or different traits), though I also found it interesting that mechanistic models do not tend to outperform simpler phenomenological alternatives, and did not gain much in the way of insights about different taxonomic groups

I think the paper is clearly written and generally achieves what it sets out to do, with a number of caveats. The main ones to me are as follows.

I'm not quite convinced of the significance of the work: I'm unaware of anything similar, and don't believe the authors overstate its importance. But I do wonder how broadly appealing it will be, given many of us fit TPCs for biological insights (e.g., comparisons of different species or contexts) without the exact model being a primary concern. I also wondered if there was a missed opportunity to identify other gaps in the kind of thermal performance data we're sampling – for example, do we only focus on adults, or Gaussian traits? Are we sampling sufficient temperatures to fit curves well (unclear from Fig 4)? With so many candidate models but narrow frame of reference biologically, the recommendations and conclusions are potentially limited in scope.

We thank the Reviewer for their thorough comments. We have now made clearer the take-home messages and the recommendations that our results allow us to make. These are outlined in comments #10 and #34, at the end of the abstract (lines 31–33), and at the “Recommendations and conclusions” section (lines 214–229, 242–271).

41. The outcomes of the conditional inference tree analysis are worrying, and aren't really delved into. What are the possible reasons for such a poor result? Some explanation, probing, or justification for the striking patterns in Fig 6 seem warranted to put this result in better context.

In our study, we show that model performance cannot be accurately predicted from sampling resolution (Fig. 4) or trait type (Fig. 5), and that there are no models that consistently yield high AICc weights (Fig. 3). These findings are consistent with the low accuracy of the conditional inference tree, given that there are no simple rules for selecting the most appropriate model for a given dataset. See also our response to comment #10 above.

42. On a similar note, I often found myself wanting more explanation and justification for decisions taken throughout the comparisons. I list these against specific line numbers below.

L39. Yes, TPCs are usually asymmetric, though Angilletta also notes that curves for survivorship during embryonic or larval stages tend to be more symmetrical. This is acknowledged for per capita fecundity on L77, but are such instances exceptions to the rule, or does the rule emerge from limited sampling of certain traits? I guess I'm querying if this point is worth emphasising earlier.

Some of the authors of the present study recently published another study on the contributions of different traits to the population growth rate of arthropods [2]. That study included TPCs of juvenile mortality, adult mortality, and fecundity, whose TPCs are commonly assumed to be symmetric. In contrast to this expectation, we found that for many species, the shape of the TPCs of these traits was asymmetric (Supplementary Figs. 8-10 of [2]). Therefore, we agree that more attention should be targeted towards understanding whether certain traits are more likely to have symmetric TPCs than others, and the underlying mechanisms for this, but this would be outside the scope of the present study.

We have now removed the sentence relating to per capita fecundity from the Introduction and replaced it with the following: “For example, consider a hypothetical trait whose TPC is strictly symmetric.”

43. L62-63. I'm curious: is number of temperatures the sole source of high resolution, or could resolution be increased by other aspects of experimental design? For example, number of inde-

pendent replicates per temperature, or clustering of temperatures around thermal optima (T_{pk}) or limits?

This led me to wonder if any insights can be gained about *how* models perform better or worse. I can think of instances where thermal optima appear to match data well, but thermal limits do not – for example, because performance was not sampled sufficiently at extremes. If researchers are interested in a certain TPC feature, should they choose one model over another to estimate that feature well?

This is a very good point and we thank the Reviewer for raising it. We have now added a paragraph in the “Recommendations and conclusions” section (lines 260–271) with our recommendations for ensuring that the data have sufficient statistical power for addressing the questions of a given study.

44. L92. Does aim 3 explicitly compare mechanistic versus phenomenological models with the same numbers of parameters? Is this important for fair comparison, or moot because model types tend to have similar complexity? Please clarify so readers can follow the logic.

The AICc weights explicitly account for the number of parameters, allowing us to objectively compare models with few or many parameters. If one wants to compare the performance of models with the same number of parameters, then one can skip the calculation of AICc weights and only examine the R^2 's. We now explain these better in Box 1 in the main text.

Note that Fig. 4 shows that there are no mechanistic models among the top 10, regardless of sampling resolution.

45. L104 and Fig 2. All Gaussian traits? At what life stages? Are there other important dimensions worth acknowledging, given this is in some ways as systematic review with value for highlighting knowledge gaps? Some of these issues are gaining attention, so it feels like an oversight worth addressing.

Given the aims of the present study, we did not collect information on life stages or other aspects besides those explicitly reported in the present study. In any case, such information would not be universally applicable to all our thermal performance datasets, such as those obtained from microbial species.

Regarding the traits analysed in this study, we classified them as physiological, emergent, or interaction traits, and not as generally producing symmetric or asymmetric TPCs. This is because our knowledge on this topic is severely lacking, as we mention in comment #42 above.

46. L122-124. Readers are told what was done but not why. Please clarify why an R^2 value of 0.5 was chosen as the relevant cut-off.

As we show in Supplementary Fig. 1, most fits included in our study had R^2 values close to 1. Thus, we arbitrarily set a cutoff of 0.5 to exclude very poor model fits. In any case, fits with very low R^2 values would yield AICc weights equal to 0, as long as there was at least one model that achieved a high R^2 against the same dataset. Choosing a different threshold would not change our qualitative conclusions. We now clarify this in Box 1.

47. L124. What is AICc weight? The per-model proportion of total AICc values? Please clarify for readers.

See our response to comment #3 above.

48. L129-135. This is the message of the paper flagged in the title. I think readers could be forgiven for wanting some biological context, rather than a list of models that performed relatively well. I realise the stated aim of the work is being delivered on, but wonder if the scope is somewhat narrow for the intended audience.

See our responses to comments #10, #23, and #34 above.

49. L135. Why focus on the top ten models in each of the scenarios in Figs 4 and 5? Are results robust to this decision? Please clarify so readers can follow the logic.

In Fig. 3, we showed that there is remarkable variation in model performance, with no model achieving consistently high performance across datasets. In particular, most cells in the heatmap of Fig. 3 have an AICc weight of ~ 0 . In light of this, our next step was to determine if differences in sampling resolution and trait type can lead to dramatic differences in model performance. It is for this reason that we chose to include only ten models per group in Figs. 4 and 5. Our results showed that neither sampling resolution nor trait type have a major effect on model performance, with the ten best-performing models per group having very low median AICc weights and very wide interquartile ranges. Including additional models would increase the complexity of the figures for effectively no gain, given that such models would necessarily have median AICc weights even closer to 0.

50. L130. Perhaps I missed it, but what should readers interpret as high AICc weights, or weights that support adequate model performance?

Please refer to our response to comment #3 above.

51. Fig 3 legend. ‘Branch lengths represent median Euclidean distances among resulting TPC fits’: it would be worth clarifying for readers how they should interpret the branch lengths of models versus the arrangement (or clustering) of models in the dendrogram.

Branch lengths represent the similarity of the fits produced by different models. For example, consider two models that descend from a common tree node. If the distance from the common node to each model is close to zero (e.g., in the case of Jöhnk and Thomas II), then this means that the two models tend to produce nearly identical fits. In contrast, if the distance from the common node to each model is large (e.g., in the case of Tomlinson-Phillips and Tomlinson-Menz), then this indicates that the two models produce similar but not fully equivalent fits.

We now provide a brief interpretation of branch lengths in the caption of Fig. 3.

52. L136. Do i), ii), and iii) correspond to (A), (B), and (C) in Fig 4? It would be worth using the same labelling if so.

That is indeed the case. We have now addressed this as suggested (lines 136–137).

53. L139-142. Again, it would help readers to know exactly what AICc weights represent, or what weights support adequate model performance. They’re mainly used to rank models, yet the point here (that the best models in all three groups had median weights of ≤ 0.05 and interquartile ranges including weights of effectively zero) seems to encourage interpretation on the absolute scale. What should we infer (here and in Fig 5) from medians being so small, when some models have weights near 1? That models tend to have little or no predictive value, or that most models are *relatively* bad and swamp high-weight models when presented this way? Likewise, what should we infer from lower medians for models fitted to higher resolution data than for models fitted to lower resolution data (or different trait groups in Fig 5)? Apologies if I’m misunderstanding, but if I was confused I suspect other interested readers will be too.

Please see our response to comment #3 above. Regarding Fig. 5, a median AICc weight close to 0 indicates that this model does not **consistently** outperform other models for a given trait type. In other words, there may be some datasets where the model happens to strongly outperform other models, but this behaviour is by no means systematic.

Regarding sampling resolution, as we mention in comment #3 above, high sampling resolution allows us to fit a wider range of models than in the case of low sampling resolution. Therefore, if no model consistently outperforms others, the AICc weights will necessarily decrease as the number of fitted models increases.

We now explain all these in Box 1.

54. L188-198. Indeed, this and patterns in Fig 6 are worrying. Was the problem explored to diagnose what could be driving it? For example, do I understand correctly that models were fitted to 29 predictors (I wasn't sure what fitting to 'different combinations of predictors' meant)? Did simpler models, models with uncorrelated predictors, or models with fewer response variables, give better prediction?

See our responses to comments #10 and #41 above. By 'different combinations of predictors', we meant that we specified all possible combinations of the 4 groups of the 29 predictor variables. These 4 groups were trait type, taxonomic information, TPC shape, and sampling resolution. The predictor combinations are shown in Fig. 6A. We have now rephrased that sentence (line 190) to avoid confusion.

55. L211-222. I guess these seem like limited take-homes for this work. Why focus on mechanistic models here if they perform no better than phenomenological ones? Fitting truly nonlinear models is challenging – even more so for non-Gaussian traits, or experimental designs that warrant mixed model approaches. Why not opt for phenomenological models, such as polynomials that can accommodate these scenarios and still allow derivation of important features (T_{min} , T_{max} and T_{pk}) in the manner of, for example, Kellermann et al 2019?

We now provide a detailed discussion of a) the advantages and disadvantages of mechanistic and phenomenological models (lines 242–259), b) model averaging (lines 222–229), and c) statistical power present in datasets (lines 260–271).

References

- [1] Rohr, J. R. *et al.* The complex drivers of thermal acclimation and breadth in ectotherms. *Ecol. Lett.* **21**, 1425–1439 (2018).
- [2] Pawar, S. *et al.* Variation in temperature of peak trait performance constrains adaptation of arthropod populations to climatic warming. *Nat. Ecol. Evol.* 1–11 (2024).

Responses to the Reviewers

Comments made by the Reviewers are in **green**, whereas our responses are in **black**.

Comments by Reviewer #1

1. I appreciate the authors' efforts to incorporate comments from reviewers. The revised version of the manuscript is much-improved and (to me) ready for acceptance.

Thank you very much!

Comments by Reviewer #2

2. Kontopoulos and colleagues have revised their manuscript in view of reviewer comments and have responded with authenticity to the feedback.

Inclusion of enzyme TPC data added depth to the analysis, and I thought the Box was a particularly effective solution to explaining the differences in model performance metrics.

I have only a few small comments remaining.

Line 65: 'parameter-rich models will systematically outperform simple alternatives, e.g., revealing two different gradients at the rising part of the TPC'. Could the models not also have divergent performance at the descending part of the TPC?

Thank you very much for your support! The shape of the TPC at its descending part may indeed vary across models, such as between those allowing for asymmetric TPCs and those enforcing symmetry before and after the peak. In any case, our goal in line 65 was to provide a single representative example of the benefits of parameter-rich models: enabling the detection of a shift in gradient at the rise of the TPC. Such a shift can be detected using the 6-parameter Sharpe-Schoolfield model and its variants. To the best of our knowledge, the existence of two distinct gradients has mainly been hypothesised for the rising part of the TPC rather than its fall. It may be possible that some traits also exhibit two gradients at the fall of their TPCs, but such a discussion would be beyond the scope of this study.

3. Line 101: What did the filtering process exclude? Why were >800 datasets excluded from the analysis?

The filtering criteria are described in Figure 2d and in the Methods section. In brief, we excluded datasets for which a) all models had R^2 values below 0.5 or b) at least one three-parameter model had an AICc value of $-\infty$.

4. Line 298: Why not negative instead of non-positive?

Certain models (e.g., Johnson-Lewin, Sharpe-Schoolfield) cannot accommodate negative or zero trait values because of their mathematical form. Therefore, to facilitate model comparisons, we removed non-positive trait values.

Comments by Reviewer #3

5. I think the authors have done a good job in revising this manuscript, and find it much improved from the version I reviewed originally. The main results remain unchanged, but the explanations of them have been made clearer, as has their broader relevance.

In particular, the addition of Box 1 (explaining the model performance metrics used) goes a long way to resolving my comments about the meaning and application of these metrics.

I especially appreciate the substantial expansion of the Recommendations and Conclusions section, which to me makes the outcomes of the work more broadly useful, in terms of developing best practices for fitting TPCs to data.

I found the authors' justifications in cases where few/no changes were made to be reasonable.

I had only one *very* minor point: should 'whereas' in L175 be 'whereby'? Perhaps I misunderstand, but 'whereas' seems to convey the opposite meaning to that intended.

We thank the Reviewer for their supporting words! We have now rephrased that sentence to avoid confusion (lines 181–183).

6. I have not executed the code, but it matches the descriptions of analyses in the manuscript. It is well-annotated and provides a README file with enough instructions for running it. I believe it is a usable resource for the community.

Thank you once again for your kind words!